# Glutathione in Protein Redox Modulation through S-Glutathionylation and S-Nitrosylation

**DOI:** 10.3390/molecules26020435

**Published:** 2021-01-15

**Authors:** Elena Kalinina, Maria Novichkova

**Affiliations:** T.T. Berezov Department of Biochemistry, Peoples’ Friendship University of Russia (RUDN University), 6 Miklukho-Maklaya Street, 117198 Moscow, Russia; novichkova-md@rudn.ru

**Keywords:** S-glutathionylation, S-nitrosylation, GSH, nitrosoglutathione, redox-regulation

## Abstract

S-glutathionylation and S-nitrosylation are reversible post-translational modifications on the cysteine thiol groups of proteins, which occur in cells under physiological conditions and oxidative/nitrosative stress both spontaneously and enzymatically. They are important for the regulation of the functional activity of proteins and intracellular processes. Connecting link and “switch” functions between S-glutathionylation and S-nitrosylation may be performed by GSNO, the generation of which depends on the GSH content, the GSH/GSSG ratio, and the cellular redox state. An important role in the regulation of these processes is played by Trx family enzymes (Trx, Grx, PDI), the activity of which is determined by the cellular redox status and depends on the GSH/GSSG ratio. In this review, we analyze data concerning the role of GSH/GSSG in the modulation of S-glutathionylation and S-nitrosylation and their relationship for the maintenance of cell viability.

## 1. Introduction

Redox-dependent processes largely determine cell viability, participating in the regulation of division, bioenergetics, and programmed death. The cellular redox status is characterized by low-molecular-weight indicators (GSH, NADH). The change in their oxidized/reduced form ratio occurs as a reaction to changes in reactive oxygen and nitrogen species (RONS) levels and, so, they can play the role of a trigger in the redox-dependent regulation of cellular processes. Undoubtedly, such an important trigger role is played by glutathione (γ-glutamyl-l-cysteinylglycine, GSH), a water-soluble tripeptide consisting of the amino acids l-glutamate, l-cysteine, and glycine, which is widely present in both eukaryotes and prokaryotes [1,2,3]. GSH is less susceptible to oxidation than Cys, which makes it the most suitable for maintaining intracellular redox status [1]. The presence of a γ-peptide bond at the Glu residue protects GSH from the action of peptidases, while the SH group at the Cys residue makes GSH a good electron donor, allowing it to participate in reactions with strong electrophiles.

Normally, the ratio of reduced glutathione (GSH) to oxidized glutathione (GSSG), GSH/GSSG—which characterizes the cellular redox status—is 100/1 in the cytoplasm, 10/1 in mitochondria, and 3/1 to 1 in the endoplasmic reticulum [4]. This ratio varies depending on the physiological state of cells, such as proliferation, differentiation, or apoptosis, and the consequences of its disturbance are significant changes in cellular signaling reactions. This role of GSH/GSSG is largely due to its regulatory effect on the functional activities of protein thiols [5,6,7].

Although Cys residues in mammalian proteins do not exceed 3% [6], they are highly sensitive to oxidative modification, which significantly affects the functioning of proteins, as thiol groups play a significant role in the formation of protein tertiary and quaternary structures and enzyme active sites. The pKa value of most SH groups of cellular proteins is more than 8.0, which keeps thiol groups predominantly protonated at physiological pH values (pH 7.0–7.4) [8,9]. However, in proteins, in the immediate vicinity of the basic amino acid residues (histidine, lysine, and arginine), the pKa of the SH groups decreases (usually to 5.0–7.0) and these thiols dissociate at physiological pH. The resulting thiolate anions (Pr-S^−^) are effective nucleophiles and have high activity with respect to electrophilic targets [10,11,12]. The reactivity of SH groups and the functional activity of proteins are largely regulated by S-glutathionylation and S-nitrosylation [6,13,14,15,16].

Under S-glutathionylation, GSH can bind to the cysteinyl residues of proteins through the creation of reversible disulfide bonds, depending on the cysteine position and redox potential [12,13]. This post-translational modification to the protein can lead to enhanced or suppressed activity, may prevent protein degradation by proteolysis or sulfhydryl overoxidation, and plays an important role in cellular signaling. At present, the dual role of S-glutathionylation in maintaining cellular homeostasis and participating in various pathological processes may be indicated [14,17]. 

Under S-nitrosylation, NO is covalently attached to the SH group of a cysteine residue and, as a consequence, can cause alterations in the cellular function of a variety of proteins [18,19]. S-nitrosoglutathione, formed as the result of GSH S-nitrosylation, serves as a NO reservoir and can transfer NO groups to new cysteine residues in transnitrosylation reactions [20,21].

The GSH/GSSG ratio can be considered a key redox sensor, which determines the redox-dependent alteration of the protein functional activity through such significant post-translational modifications as S-glutathionylation and S-nitrosylation. The modulation of the activity of these reactions in response to a change in the GSH/GSSG ratio, as a result of an increase or decrease in RONS levels, provides a significant contribution to cell functional adaptation to redox changes of the environment [22,23,24,25]. 

In this review, we analyze data concerning the roles of GSH and GSH/GSSG in the redox modulation of S-glutathionylation and S-nitrosylation and their relationship for the maintenance of cell viability.

## 2. Glutathione and Protein S-Glutathionylation

A widespread form of cysteine modification is S-glutathionylation—the reversible formation of protein mixed disulfides with GSH (Pr-SSG)—which occurs in the cell under physiological conditions and oxidative stress, both spontaneously and enzymatically. The S-glutathionylation of proteins suggests the possible involvement of this post-translational modification in cellular signaling and the redox regulation of protein functions [6,13,24]. In addition to the potential regulatory role, S-glutathionylation can serve as a means of GSH storage, as well as protection from the irreversible oxidation of protein thiol groups under stress conditions [26], often due to a temporary loss of primary protein activity as, if the modified sulfhydryl group of a protein is functionally critical, S-glutathionylation can render the protein inactive or alter its activity, ultimately disrupting cellular functions [27]. In addition, this reaction can affect a change in conformation and/or charge, which can modify the function of the protein, as the attachment of GSH introduces an additional negative charge at the expense of the glutamic acid residue.

Non-enzymatic S-glutathionylation reactions can occur during thiol-disulfide exchange, through the participation of protein thiol (Pr-SH) and oxidized glutathione GSSG:Pr-SH + GSSG → Pr-SSG + GSH.

The equilibrium constant of the reaction K_mix_ is expressed by the ratio [Pr-SSG]·[GSH]/[Pr-SH]·[GSSG], where the extent of S-glutathionylated proteins ([Pr-SSG]/[Pr-SH]) strongly depends on the local ratio [GSH]/[GSSG] [28,29]. The S-glutathionylation of most proteins with typical redox potential (K_mix_~1) by about 50% can occur when this ratio drops very dramatically (i.e., from 100:1 to 1:1). These extreme conditions are rare in vivo. Therefore, for most proteins, the spontaneous formation of Pr-SSG—as a result of the exchange of Pr-SH and GSSG—is not common and, as a rule, takes place under pathological conditions [30]. In addition, a variant of thiol-disulfide exchange between Pr-SH and a protein which is already S-glutathionylated (Pr′-SSG) is possible:Pr-SH + Pr’-SSG → Pr-SSG + Pr’-SH.

S-glutathionylation can occur when Pr-SH or GSH reacts with an oxidized derivative of the protein cysteine residue; for example, sulfenic acid (-SOH), thiyl radical (-S^•^), or S-nitrosyl (-SNO) group. Thus, when Pr-SH is oxidized with, for example, H_2_O_2_, sulfenic acid (Pr-SOH) is formed and then quickly reacts with GSH to form Pr-SSG:$$\text{Pr-SOH}\;+\;\mathrm{GSH}\;\to {\;\text{Pr-SSG}\;+\;H}_{2}O.$$

Sulfenic acid is unstable and can undergo further oxidation to sulfinic acid (Pr-SO_2_H) and eventually to sulfonic acid (Pr-SO_3_H), the formation of which, as a rule, leads to the irreversible deactivation of the protein. Thus, the S-glutathionylation of sulfenic acid can prevent the oxidation of protein cysteine residues [6,25]. Under physiological conditions, the intracellular level of H_2_O_2_ is in the sub-micromolar range (10^−9^–10^−7^ M) [31]. Therefore, in vivo spontaneous S-glutathionylation proceeds rather slowly by this mechanism. 

The formation of S-glutathionylated protein is also possible, due to its interaction with GSSG in the form of sulfenic acid (GSOH):$$\text{Pr-SH}\;+\;\mathrm{GSOH}\;\to {\;\text{Pr-SSG}\;+\;H}_{2}O.$$

Thus, it is obvious that the S-glutathionylation of proteins can occur spontaneously; however, the rate and extent of this process increases with the participation of enzymes, among which glutathione transferase isoform P1-1 (GSTP1-1) plays the greatest role [32,33]. GSTP1-1 has been shown to facilitate S-glutathionylation for a number of proteins, including peroxiredoxin 6 (Prx6) [34,35], aldose reductase [36], actin [37], histone H3 [38], 5’AMP-activated protein kinase (AMPK) [32], estrogen receptor α [39], heat shock protein BiP, protein disulfide isomerase (PDI), calnexin, calreticulin, and sarcoplasmic reticulum Ca^2+^-ATPase (SERCA) [33].

During the reaction, GSTP1-1 binds GSH in the active center and decreases the pKa of the GSH cysteine residue from 9.2 to 6.3 [40], deprotonates it with the participation of Tyr7, forming a thiolate anion (GS^−^), which is transferred to the cysteine residue in the substrate. In cells with the Tyr7 GSTP1-1 mutation, a decrease in the total content of S-glutathionylated proteins has been observed upon treatment with GSSG, the mimetic of oxidized glutathione NOV-002 and diazeniumdiolate-based NO-donor prodrug PABA/NO [36]. An example is the S-glutathionylation of Prx6 from the 1-Cys groups of peroxiredoxins. As a result of human Prx6 peroxidase activity, the Cys47 in the active center is oxidized to sulfenic acid; this deprives it of activity; as for the reduction of which, a second thiol is required to form a mixed disulfide, then a sulfhydryl group. However, the availability of the sulfenic group is low, due to the peculiarities of the globular structure of Prx6. Prx6 activation occurs during the formation of a heterodimer with GSTP1-1, which promotes the S-glutathionylation of Cys47 Prx6. The conformational changes of the heterodimer occur, providing the formation of a disulfide bond between Cys47 GSTP1-1 and Cys47 Prx6, followed by the reduction of disulfide with the participation of GSH and the regeneration of Cys47 Prx6 [34].

The enzymatic S-glutathionylation of AMP-activated protein kinase (AMPK) occurs not only with the participation of GSTP1-1, but also GSTM1-1 in the absence of strong oxidants (i.e., under conditions which are similar to physiological oxidation). S-glutathionylation occurs at the Cys299 and Cys304 residues and causes conformational changes that activate the kinase activity of human AMPK [32].

The ability of S-glutathionylation was found in the enzyme glyoxylase 2 (Glo2). Glo2 hydrolyzes S-d-lactoylglutathione to glutathione and lactic acid, while GS^−^ is formed in the active center of Glo2, similar to GSTP1-1 [41]. It has been established that actin and malate dehydrogenase can serve as substrates for S-glutathionylation by Glo2 [42].

S-glutathionylation is a reversible post-translational modification and, as a rule, deglutathionylation proceeds with the participation of enzymes and is more carefully regulated, in comparison with S-glutathionylation. Glutaredoxin (Grx) is one of the most effective and well-studied enzymes, which reduces Pr-SSG. In the traditional classification, they are divided into mono- and dithiol Grx, depending on whether one or two cysteine residues, respectively, are in the active center. The role of dithiol Grxs is mainly considered in the regulation of reversible S-glutathionylation [26,43,44]. 

Mammalian dithiol Grxs, Grx1 and Grx2, are found in many cellular compartments; however, Grx1 is mainly present in the cytosol (~1 μM) and mitochondrial intermembrane space (~0.1 μM), while Grx2 is localized mainly in the mitochondrial matrix (~1 μM) [45,46]. Being thiol oxidoreductases, Grx1 and Grx2 contain the CXXC motif (Cys^N^-XX-Cys^C^; CPYC in Grx1 and CSYC in Grx2) in the active site. In addition, they are characterized by the presence of a thioredoxin fold, consisting of four β-sheets surrounded by three α-helices, and a site responsible for stabilizing GSH. Grx uses GSH as a co-substrate for the reduction of Pr-SSG mixed disulfides.

It should be noted that, depending on the value of the GSH/GSSG ratio, Grx can not only carry out deglutathionylation but, on the contrary, may promote S-glutathionylation (Figure 1). Grx2 functions as a glutathionylation enzyme under a decrease of GSH/GSSG and an increase in the level of Н_2_О_2_ (e.g., in relation to the respiratory complex I) whereas, at a high level of GSH/GSSG, and low concentrations of Н_2_О_2_, Grx2 has a deglutathionylating activity [47]. The putative mechanism of S-glutathionylation proceeds in several stages: first, there is a nucleophilic attack of the disulfide bond GSSG by the thiolate anion Grx-S^−^, along with the formation of the glutathionylated intermediate Grx-SSG, from which the activated cationic radical [GS^●^]^+^ is transferred to the target protein with the formation of Pr-SSG, while Grx is again capable of catalyzing the reaction. For this process, the possibility of the reversible formation of Grx-S_2_ from Grx-SSG is also noted [48].

In addition to Grx, the ability to catalyze deglutathionylation has been observed in some other enzymes (Table 1).

The isoform of glutathione transferase, GSTO1-1, has the ability to catalyze protein deglutathionylation [54,55]. The isozyme is structurally similar to Grx, including Trx-like folding and a glutathione binding site, where it can form a disulfide bond with a conserved cysteine residue in the active site [53]. Other GST isoforms, including GSTA, GSTM, GSTP, GSTT, GSTS, and GSTZ, in contrast, have catalytic tyrosine or serine residues. In addition, GSTO1-1 has a relatively accessible pocket in the active site, which can potentially accommodate a protein or peptide as a substrate [54,64]. GSTO1-1 catalyzes Grx-like protein deglutathionylation in two similar stages: in the first, the Cys32 of the active site in human GSTO1-1 interacts with Pr-SSG, resulting in reduced Pr-SH and mixed disulfide GSTO1-1-Cys_32_S-SG, which is deglutathionylated with the participation of GSH to form GSSG and functional active GSTO1-1, which is capable of catalyzing the deglutathionylation of the next protein substrate [54]. The question of the role that GSTO1-1 plays in the S-glutathionylation of proteins remains open [53,55].

The process of deglutionylation is also carried out with the participation of the main members of the Trx family, which play essential roles in maintaining cellular redox homeostasis. Thioredoxins (Trx) 1 and 2 restore disulfide bonds in proteins. This process involves two cysteine residues of the Trx active site (Cys-X-X-Cys), where the disulfide bond is transferred from the substrate protein to Trx. Then, the oxidized Trx is reduced by the NADPH-dependent Trx reductase (TrxR) [65]. In addition, using the mechanism of dithiol reduction, Trxs are able to carry out deglutathionylation without the participation of GSH, which has been shown for glyceraldehyde-3-phosphate dehydrogenase, Prx3, 20S proteasome, and NOS3 [59,60,61,62]; however, the exact mechanism of deglutathionylation has not yet been determined.

The ability to deglutathionate proteins has also been observed in protein disulfide isomerases (PDIs), which are also included in the Trx family [63]. However, the significance of the PDI contribution to this process is not yet clear, as their main function is the exchange of the disulfide bonds of PDIs and target proteins. PDIs are enzymes of the endoplasmic reticulum, which are specifically responsible for protein folding through the oxidation of newly formed proteins and isomerization of proteins with improperly formed disulfide bonds, achieving the formation of their native structure. Moreover, PDIs can be secreted by the cell or associated with the cell surface to maintain proteins in a reduced state [62].

Sulfiredoxin (Srx), for which the ability to reduce the cysteine residue oxidized to sulfinic acid in the active site of typical 2-Cys perxiredoxins (2-Cys Prx) was originally established, is also capable of deglutathionylation, in relation to at least the Prx isoforms [13,56,57], actin, and tyrosine protein phosphatase 1B [58]. For example, it has been shown in vitro that human Prx1 can be S-glutathionylated at three out of four cysteine residues—Cys52, Cys173, and Cys83—and deglutathionylation at Cys83 and Cys173 is catalyzed by Srx, while deglutathionylation at Cys52 is carried out by Grx [56]. The mechanism of Srx-catalyzed deglutathionylation has not yet been fully determined. The data indicate that it proceeds by a mechanism similar to that catalyzed by Grx through the formation of the Srx-SSG intermediate glutathionylated at the conservative Cys99 residue [56]. Srx-catalyzed deglutathionylation appears to have broad substrate specificity. In HEK293 cells transfected with Srx, a decrease in the total content of S-glutathionylated proteins formed under conditions of nitrosative stress after treatment with the nitric oxide donor PABA/NO has been demonstrated [58].

## 3. Protein S-Nitrosylation and Denitrosylation 

S-nitrosylation, as well as S-glutathionylation, serves as a reversible post-translational modification of thiol groups of proteins [15,66,67]. In mammals, nitric oxide (NO) is mainly synthesized by nitric oxide synthase (NOS), for which three isoforms are known: Two constitutive—NOS1 (neuronal, nNOS) and NOS3 (endothelial eNOS)—as well as inducible NOS2 (iNOS), which have approximately 50% homology [68,69,70]. The activities of constitutive NOS1 and NOS3 are mainly regulated by phosphorylation, S-nitrosylation, protein–protein interactions, and changes in calcium levels, due to which steady-state NO concentrations are largely maintained [71]. On the contrary, inducible NOS2 produces high levels of NO in response to various factors [71]. In addition, the isoform mtNOS, a homolog of NOS1, has been found in the inner mitochondrial membrane and matrix. Its influence on the mitochondrial function has been intensively studied in recent years [72,73]. 

Under conditions of oxidative stress (NOS3 incubated with 2 mM GSSG, molar ratio of NOS3 to GSSG of 1:250), S-glutathionylation causes the decoupling of the NOS3 function and switches its activity from NO synthesis to the generation of $O_{2}^{•-}$, thereby affecting the regulation of vascular tone [74,75]. In the human NOS3 reductase domain, the sites of S-glutathionylation are Cys689 and Cys908. Their modification in the presence of high concentrations of GSSG led to a noticeable increase in the formation of $O_{2}^{•-}$, as shown in experiments carried out on the purified enzyme, as well as endothelial cells and intact vessels [75]. In the presence of GSH (1 mM), Grx1 reversed GSSG-mediated S-glutathionylation of NOS3, facilitating restoration of NO-synthase activity [75]. However, an increase in [GSSG]/[GSH] above 0.2, which can be observed in tissues under ischemia-reperfusion, causes S-glutathionylation at the Cys382 in the NOS3 oxygenase domain with the participation of Grx1. On the contrary, a decrease in [GSSG]/[GSH] below 0.1 leads to the deglutathionylation of the site. Thus, the S-glutathionylation of NOS3 by Grx1 at Cys382 is sensitive to fluctuations in the [GSSG]/[GSH] level and provides a unique mechanism for protection of the NOS3 thiol against oxidation.

The formed NO is actively involved in signal transduction, either as an activator of guanylate cyclase—which synthesizes cGMP as a secondary messenger—or due to post-translational modifications of biomolecules, which occurs with the participation of NO itself or under the action of NO derivatives and includes the S-nitrosylation of protein thiols, the nitrosylation of transition metal ions, and the oxidative nitration of various molecules, such as tyrosine residues, amines, fatty acids, and guanine [76,77,78].

As a rule, for the S-nitrosylation of cysteine thiol groups, one-electron oxidation is necessary, which occurs with the participation of О_2_ or a transition metal ion (e.g., iron or copper) [76]. The variant of the possible direct addition of NO occurs quite rarely, with the participation of a thiyl radical: $${\mathrm{NO}\;+\;\text{Pr-S}}^{•}\;\to \;\text{Pr-SNO}.$$

When NO interacts with O_2_, a set of oxides with a higher degree of nitrogen oxidation (so-called auto-oxidation) are formed, among which N_2_O_3_ is considered as the main nitrosylating agent that promotes the appearance of nitrosothiol and nitrite: $${2\mathrm{NO}\;+\;O}_{2}\;\to {\;2\mathrm{NO}}_{2},\;{\mathrm{NO}\;+\;\mathrm{NO}}_{2}\;\to {\;N}_{2}O_{3},\;N_{2}O_{3}\;+\;\text{Pr-SH}\;\to {\;\text{Pr-SNO}\;+\;\mathrm{NO}}_{2}^{-}\;{+\;H}^{+}.$$

The interaction of NO_2_ with thiol to form a thiyl radical and its further reaction with NO is possible: $${\mathrm{NO}}_{2}\;+\;\text{Pr-SH}\;\to {\;\text{Pr-S}}^{•}{\;+\;\mathrm{NO}}_{2}^{-}\;{+\;H}^{+},\;{\mathrm{NO}\;+\;\text{Pr-S}}^{•}\;\to \;\text{Pr-SNO}.$$

Both processes are limited by the speed of NO_2_ formation and its availability. The formation of GSNO can occur in the same way.

A variant of S-nitrosylation catalyzed by transition metal ions (Fe^3+^ or Cu^2+^) has been described; in this case, one-electron oxidation of NO occurs and the resulting nitrosonium ion (NO^+^) can nitrosylate a thiol located in its immediate vicinity: $${\mathrm{Me}}^{\left(n+1\right)}\;+\;\mathrm{NO}\;\to {\;\mathrm{Me}}^{n}-{\mathrm{NO}}^{+},\;{\mathrm{Me}}^{n}-{\mathrm{NO}}^{+}\;+\;\text{Pr-SH}\;\to {\;\text{Pr-SNO}\;+\;\mathrm{Me}}^{n}{+\;H}^{+}.$$

This mechanism takes place during the autonitrosylation of hemoglobin and the formation of nitrosoglutathione (GSNO), through the participation of ceruloplasmin and cytochrome c [76,79].

The S-nitrosylation of proteins can be accomplished by transnitrosylation, by which a low-molecular-weight nitrosothiol (e.g., GSNO), or a protein nitrosylated at a cysteine residue or containing a nitrosylated metal ion (e.g., in heme), interacts with the protein and transfers NO to the protein cysteine residue. This transfer between thiol-containing compounds promotes the sequential remote transfer of NO from the site of its synthesis, including NO transfer between different sub-cellular structures [80,81]. 

Trans-nitrosylation can occur between Cys residues (Cys-to-Cys) or between a metal and Cys residues (Me-to-Cys): $${\text{Pr-S}}^{-}\;+\;\text{Pr’-SNO}\;\to {\;\text{Pr-SNO}\;+\;\text{Pr’-S}}^{-},\;{\text{Pr-S}}^{-}\;+\;\mathrm{Me-NO}\;\to \;\text{Pr-SNO}\;+\;\mathrm{Me}.$$

The terms “S-nitrosylase” and “trans-nitrosylase” are used to designate the enzymes involved in the transfer of NO groups [79]. In a process with the participation of a transition metal ion (e.g., iron or copper), the NO group can be transported intra- or inter-molecularly. For example, in hemoglobin, the NO group is transferred from the heme iron to neighboring thiols in the same molecule; whereas, in cytochrome *c*, NO is coordinated with the iron atom and transferred to the SH group of glutathione to form GSNO [81]. In Me-to-Cys trans-nitrosylases, the transition metal can perform the redox functions that are necessary for the formation of Pr-SNO without molecular oxygen [82].

During trans-nitrosylation, the thiolate anion of the recipient carries out a nucleophilic attack of the nitrogen in the nitrosyl group of the donor [83]. To date, several trans-nitrosylases have been identified (Table 2), for which only specific cysteines seem to be targets that allow for the selective regulation of certain cellular signaling pathways [80].

The main factor determining the selective transfer of NO is the physical distance between the donor (S-nitrosylase) and recipient thiol group. As a rule, the nitrosylated protein contains the I/L-X-C-X2-D/E motif. Another factor is the redox potential between thiols. Thus, trans-nitrosylation occurs only when two proteins interact directly and have corresponding redox potentials that provide electron transfer with subsequent NO transfer. It is assumed that the physical association of the two proteins causes their conformational change, allowing the recipient thiol to form a thiolate anion, which then attacks the donor’s nitrosyl group [80].

An interesting example of an enzyme involved in trans-nitrosylation is glyceraldehyde-3-phosphate dehydrogenase (GAPDH). Human GAPDH is S-nitrosylated at the catalytic residue Cys152 (in humans) using calcium- and zinc-dependent proteins (participants in the process of inflammation and immune response), nitrosylated in turn under the action of NOS2, which is accompanied by a loss of pro-inflammatory properties. S-nitrosylated GAPDH (GAPDH-SNO) interacts with the E3 ubiquitin-protein ligase Siah1 and moves from cytoplasm to nucleus. In the nucleus, Siah1 initiates ubiquitination and the degradation of nuclear proteins to initiate apoptosis, while GAPDH-SNO binds to p53, which also activates apoptosis. In addition, GAPDH-SNO transnitrosylates proteins involved in DNA transcription and repair. These proteins include SIRT1 and HDAC2 deacetylases, which are inhibited after S-nitrosylation. On the contrary, DNA-activated protein kinase (DNA-PK), which is involved in DNA repair, is activated by S-nitrosylation with the participation of GAPDH-SNO [88,97]. In response to stress, GAPDH is translocated into the mitochondria, where it transnitrosylates Hsp60, acetyl-CoA thiolase, and VDAC1, thus affecting membrane permeability, the regulation of mitochondrial function, and cell death [98,99].

S-nitrosylation is a reversible post-translational modification; hence, the removal of the NO group is an important aspect of signal transduction involved with it. It has generally been accepted that the levels of cellular nitrosothiols are low, due to the high reversibility of the process. Initial information on denitrosylation has indicated that it is an unregulated and spontaneous reaction. Several non-enzymatic mechanisms of denitrosylation that can potentially act in vivo have been described. These include reactions involving transition metal ions, nucleophilic compounds, and reducing agents, such as glutathione, ascorbate, bilirubin, and sulfite [100,101].

The most S-nitrosylated proteins are rapidly denitrosylated by reducing agents, such as GSH. However, there are targets for NO that are capable of forming stable S-nitrosothiols in vivo, where stable nitrosothiols appear as a result of the protein conformational changes that reduce the availability of the NO group in the solution [102]. Some of these proteins are the generally recognized targets of NO. It has been suggested that physiological signaling involving NO uses exactly stable S-nitrosothiols, for which highly specific enzymatic intracellular denitrosylation pathways are probably used to complete NO-dependent signaling, involving the cysteine residues of proteins. A large amount of data has shown that the denitrosylation process is catalyzed by the several enzymes, both in vitro and in vivo (see Table 3) [103,104,105].

The main denitrosylases are S-nitrosoglutathione reductase (GSNOR) and Trx, for which the role has been shown in vivo [104,106]. In addition, there are in vitro data obtained in cell lysates or isolated systems on the denitrosylase activity of carbonyl reductase, xanthine oxidase, Cu/Zn superoxide dismutase, PDI, glutathione peroxidase, and glutaredoxin [103,105]. The reaction products (depending on the enzyme) are various S-nitrosothiols, NO, peroxynitrite, hydroxylamine, and ammonia.

The most important denitrosylase is GSNOR. Although GSNOR is formally characterized as a class III alcohol dehydrogenase (ADH5) or glutathione-dependent formaldehyde dehydrogenase (FDH), it is most active against the GSNO substrate as a reductase and does not use other S-nitrosothiols as a substrate [107]. GSNOR is found in most human tissues, but its greatest activity has been observed in the liver [108]. The maximum content of GSNOR has been found in the cytoplasm, but this enzyme was also found in the nucleus [109].

GSNOR is active as a homodimer. The recovery of GSNO occurs during an irreversible reaction, the products of which are not involved in the S-nitrosylation of cellular proteins. In the first step, GSNO is reduced to the unstable intermediate *N*-hydroxysulfinamide (GSNHOH), using NADH as a specific co-substrate. At the next stage of the reaction, depending on the local concentration of GSH, *N*-hydroxysulfinamide is either decomposed to hydroxylamine and GSSG under physiological GSH levels in millimolar range or, under low GSH levels, spontaneously converts to glutathione sulfinamid, which can be hydrolyzed to glutathione sulfinic acid and ammonia [132]. This has been shown through in vitro experiments where low concentrations of GSH (0–1 mM) promoted the rearrangement of GSNHOH to glutathione sulfinamide, then into sulfinic acid and ammonia; however, the presence of 5 mM GSH favored yields of hydroxylamine and GSSG [133]. 

The Trx-dependent system has a wide range of denitrosylation substrates [93,134]. The reaction catalyzed by Trx involves its direct interaction with Pr-SNO (Figure 2). The attack of the more nucleophilic cysteine residue in the active site (i.e., Cys32 of human Trx1) can occur either against the sulfur atom or the nitrogen atom of the S-nitroso group of the nitrosylated protein. In the first case, it leads to the formation of a mixed disulfide between Trx and the S-nitrosylated protein with the release of the nitroxyl anion (NO^−^); then, due to the second cysteine of the Trx active center (i.e., Cys35 of human Trx1), the thiol group of the protein is reduced, while the intermolecular disulfide bond is replaced by the intramolecular bond in the active center of thioredoxin (Trx-S_2_). In the second case, at the beginning, a trans-nitrosylation reaction occurs with the formation of denitrosylated protein (Pr-SH) and S-nitrosylated Trx-NO, followed by the release of the nitrosyl group (in the form of NO or NO^−^) and the formation of an intramolecular disulfide bond in thioredoxin, which is reduced by TrxR with the participation of NADPH [93,119]. It is assumed that the geometry and electronic structure near the ONS group of the substrate protein may be responsible for the choice of the attack site; on the other hand, it is possible that structural elements near the Trx active center play a decisive role [119].

A wide range of substrates have been identified for denitrosylation catalyzed by Trx. For example, cytosolic caspase 3 and caspase 8 are activated by the denitrosylation action of Trx1, while mitochondrial caspase 3 is denitrosylated by Trx2 upon activation of Fas-induced apoptosis [93,120,135]. Trx also denitrosylates NF-κB after stimulation with cytokines, illustrating the importance of denitrosylation for immune signaling [124].

It is likely that Trx catalyzes the denitrosylation of participants in signaling cascades activated by insulin in adipocytes, which is especially important as NOS2 is induced in obesity [122].

Most cells express various Trx-like proteins; however, there is currently insufficient information on the roles these proteins play in denitrosylation processes. However, it has been found that protein TXNDC17 (TRP14/TXNL5), a representative of a large group of proteins with Trx-like domains, is capable of denitrosylating GSNO [136]. 

It has recently been found that Grx, which is also a member of the Trx superfamily proteins, may possess denitrosylase activity. It has been shown that the low-molecular-weight S-nitrosothiols L-Cys-SNO and GSNO are substrates of Grx1 while, among the proteins, caspase 3 and cathepsin B are dentrosylated by this cytosolic isoform of glutaredoxin [103]. As caspase 3 and cathepsin B play important roles in the development of apoptosis, the functional activity of Grx1 in the regulation of apoptosis may be carried out through their denitrosylation. As the active centers of Grx dithiol isoforms, similar to Trxs isoforms, contain two cysteine residues CXXC (Cys^N^-XX-Cys^C^; CPYC in Grx1 and CSYC in Grx2), it has been assumed that denitrosylation, in this case, proceeds in a similar manner as Trx, with the formation of an intramolecular disulfide in the active center of glutaredoxin. The reduced dithiol Grxs catalyze the denitrosylation reaction in the absence of GSH, whereas monothiol Grx only does so in the presence of GSH [103].

Another member of the Trx family, PDI, which is localized on the cytoplasmic membrane, promotes NO transfer from extracellular Pr-SNO to intracellular thiols, suggesting that PDI is involved in NO transfer from the extracellular environment to the cytoplasm [128]. However, it remains unclear whether this function is mediated directly by PDI or by an alternative membrane protein whose redox state is regulated by PDI. It has been found that PDI catalyzes the degradation of GSNO in vitro. This reaction apparently proceeds with the participation of cysteine residues in the active centers of two PDI sub-units through several intermediates with the formation of oxidized PDI and NO [129]. The involvement of PDI in NO transfer from extracellular albumin to intracellular metallothionein has been shown [130].

## 4. Nitrosoglutathione: Its Formation and Role in the Relationship and Control of S-Glutathionylation and S-Nitrosylation

GSNO is the most abundant low-molecular-weight S-nitrosothiol and an important NO donor. It is present in both animals and plants. The efficient and simple chemical synthesis of GSNO can be carried out by the interaction of GSH and nitrous acid [137]: $${\mathrm{GSH}\;+\;\mathrm{HNO}}_{2}\;\to {\;\mathrm{GSNO}\;+\;H}_{2}O.$$

However, it should be noted that the exact mechanisms leading to the formation of GSNO in vivo remain unclear. Apparently, one of the possible mechanisms for the synthesis of GSNO is the interaction of GSH with N_2_O_3_, which is formed due to the auto-oxidation of NO [138].

The appearance of GSNO is possible when NO interacts with the thiyl radical of glutathione GS^•^, which is formed as another product of NO auto-oxidation [138]:$${\mathrm{NO}}_{2}\;+\;\mathrm{GSH}\;\to {\;\mathrm{GS}}^{•}{\;+\;H}^{+}{\;+\;\mathrm{NO}}_{2}^{-},\;\;{\mathrm{GS}}^{•}\;+\;\mathrm{NO}\;\to \;\mathrm{GSNO}.$$

As a result of the interaction of NO with the superoxide radical anion, peroxynitrite is formed; the protonated form of which can decompose, with the formation of nitric oxide (NO_2_) and hydroxyl radical (^•^OH), leading to the further formation of GS^•^: NO + O2•− + H+ → ONOOH,ONOOH → NO2 + OH•, OH• + GSH → GS• + H2O,NO2 + GSH → GS• + NO2− + H+,GS• + NO → GSNO.

Despite the fact that many researchers do not believe that there is a direct reaction between thiols and NO, the formation of GSNO from GSH and NO in the presence of an electron acceptor (it may be O_2_) has been assumed [139] and was proven for sub-millimolar NO concentrations (<0.6 μM) [138]. It has been assumed that, at first, GSH can interact with free NO to form the hydroxyl amino radical (GSN^•^OH); then, an electron acceptor (e.g., O_2_) can convert it to GSNO [139]: NO + GSH → GSNOH•,GSNOH• + O2 → GSNO + H+ + O2•−.

GNSO formed in mitochondria can translocate to various parts of the cell, where it can participate in the transnitrosylation reactions of a number of proteins, such as NF-κB, STAT3, AKT, EGFR, and IGF-1R, the S-nitrosylation of which significantly affects their activity [79,140].

In recent years, interest has increased in the study of nitro derivatives of high unsaturated fatty acids (NO_2_-FAs) which, by themselves or as part of complex lipids, exhibit a rather wide range of biological functions in humans, animals, and plants and are considered to be signal transducers. Although the in vivo mechanisms of formation of NO_2_-FAs are not yet fully understood, it has been shown that they proceed with the participation of NO and its derivatives. On the other hand, NO_2_-FAs have been shown to be capable of releasing NO and carrying out post-translational protein modifications through nitroalkylation [141,142,143,144], due to which NO_2_-FAs (e.g., nitro-oleic and nitro-linoleic acids) are able to provide vasodilating, antioxidant, and anti-inflammatory action, which are important for the physiology of animals and plants, promoting the activation of defense mechanisms in stressful conditions. 

For plants—in particular, for Arabidopsis—it has been shown that nitrolinolenic acid (NO_2_-Ln) can release NO at physiological pH and temperature [142]. This NO may contribute to the generation of intracellular GSNO. It has been assumed that the direct interaction of NO and GSH is possible in Arabidopsis, due to the sub-micromolar concentration of NO required for GSNO synthesis [138], which has been demonstrated to be produced from NO_2_-Ln (0.21 μM/min) [142,145].

It is important to note that, under interaction with protein thiols, GSNO can lead to the S-glutathionylation of proteins, with the formation of a mixed thiol and nitroxyl [49,76]:Pr-SH + GSNO → Pr-SNO + GSH,Pr-SH + GSNO → Pr-SSG + HNO.

A possible variant is the reaction of trans-nitrosylation with the formation of GSNO during the interaction of Pr-SNO with GSH, while GSH takes on the NO group with the formation of GSNO [146]:Pr-SNO + GSH → Pr-SH + GSNO.

GSH-mediated trans-nitrosylation is more preferable than S-glutathionylation [147], which is possibly due to the fact that the nitrogen atom in the S-N bond is more positive than the sulfur atom and, thus, it is more favorable for the nucleophilic attack of GSH [148].

Due to the change in the GSNO level, the regulation of cell sensitivity to the activation of apoptosis can be carried out. In several HNSCC lines of human head and neck squamous cell carcinoma, it has been shown that the activation of STAT3 by phosphorylation was reversibly inhibited by GSNO, due to the S-nitrosylation of human STAT3 at the Cys259 residue [149]. In addition, GSNO contributed to a decrease in basal and cytokine-stimulated activation of NF-κB in HNSCC cells. The decreases in STAT3 and NF-κB activity upon treatment with GSNO correlated with a decrease in the proliferation and activation of apoptosis of HNSCC cells. Through in vivo model experiments in mice with human HNSCC tumor xenografts, tumor growth was reduced by systemic treatment with GSNO and further decreased in combination with cisplatin chemotherapy and radiation therapy. Thus, the potential possibility of using GSNO to block NF-κB and STAT3, which are responsible for cell survival and proliferation, has been suggested to enhance the therapeutic effect of traditional treatment methods [149].

As a NO donor, GSNO can induce apoptosis through the S-nitrosylation of Prx2. For example, in cells of non-small cell lung carcinoma, the S-nitrosylation of human Prx2 at Cys51 and Cys172 caused by GSNO disrupted the formation of the Prx2 dimer and suppressed its antioxidant activity, causing the accumulation of endogenous H_2_O_2_ and the activation of AMP-activated protein kinase (AMPK), which then activated sirtuin 1 (SIRT1). Phosphorylated SIRT1 loses its deacetylase activity against p53 in A549 cells or FOXO1 in NCI-H1299 cells, which ultimately leads to the apoptosis of tumor cells [150].

In vitro studies have shown that proteins, such as papain, creatine phosphokinase, and glyceraldehyde 3-phosphate dehydrogenase are sensitive to both S-nitrosylation and S-glutathionylation by GSNO, while alcohol dehydrogenase, bovine serum albumin, and actin apparently become S-nitrosylated [147]. In addition, the treatment of cells with PABA/NO [*O*^2^-{2,4-dinitro-5-[4-(*N*-methylamino)benzoyloxy]phenyl} 1-(*N*,*N*-dimethylamino) diazen-1-ium-1,2-diolate] led to a dose-dependent increase in intracellular NO followed by S-nitrosylation, the level of which was extremely low. However, a very high level of S-glutathionylation was detected for some proteins, including β-lactate dehydrogenase, Rho GDP-dissociation inhibitor β, ATP synthase, elongation factor 2, PDI, nucleophosmin-1, actin, protein tyrosine phosphatase 1B, and glucosidase II [151,152]. These data indicate that there may be two different pools of S-nitrosylated proteins, one of which remains stable, while the other is labile with respect to GSH and is subject to the rapid conversion to S-glutathionylated products. However, the conditions that favor the formation of Pr-SSG, in comparison with Pr-SNO, have not yet been determined [13].

The fact that S-nitrosylation of cysteine residues can serve as a transient intermediate has been evidenced by studies on rat smooth muscle cells treated with CysNO. The total pool of reduced thiols did not change significantly, while the level of reduced non-protein thiols—in particular, GSH—decreased. A decrease in GSH content prior to the use of CysNO by pre-treatment with BSO, which is an inhibitor of gamma-glutamylcysteine synthetase (a key enzyme of GSH de novo synthesis), led to a significant increase in S-nitrosylated proteins, indicating a violation of the ability to reduce Pr-SNO due to S-glutathionylation [148]. In contrast, the increase in cellular GSH by the GSH ethyl ester after CysNO treatment significantly attenuated the S-nitrosation of proteins. In addition, the treatment of cells with only one selective GSNOR inhibitor N6022 did not change the S-nitrosylation pattern; whereas, in the treatment with CysNO, the inhibition of GSNOR significantly increased the Pr-SNO level. These data illustrate that interventions that lead to a decrease in cellular redox and repair status can lead to imbalances in the levels of S-nitrosylated proteins [153].

## 5. Conclusions

Cellular redox status, as the state of oxidant/antioxidant balance, largely determines the viability of cells, including the processes of proliferation, differentiation, bioenergetics, and apoptosis. To a large extent, this is facilitated by the redox-dependent post-translational modifications of proteins, which alter their functional activity. Among these reactions, S-glutathionylation and S-nitrosylation are the best studied, although their relationship and regulation are still questionable. The function of a connecting link and a "switch" between S-glutathionylation and S-nitrosylation can be performed by GSNO, the level of which depends on the GSH content, the GSH/GSSG ratio, and RONS levels.

The algorithms that determine the ratio of S-glutathionylation and S-nitrosylation will be the subject of further study; however, we can already say that they are determined, on one hand, by the RONS ratio and, on the other, by GSH/GSSG and largely depend on the conditions that maintain the intracellular level of GSH, including its de novo synthesis and the rate of GSSG recovery. Trx family enzymes (Trx, Grx, PDI) play an important role in the regulation of these processes, the activity of which is determined by the cellular redox status and the GSH/GSSG level.

Taken together, this allows us to emphasize the important role of GSH in protein redox modulation—through S-glutathionylation and S-nitrosylation—in order to determine the contribution of these processes to the maintenance and regulation of cellular redox homeostasis.

## Figures and Tables

**Figure 1 molecules-26-00435-f001:**
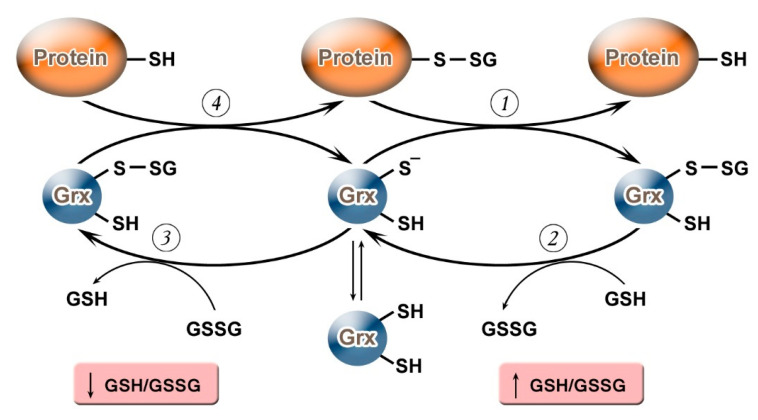
Glutaredoxin catalytic mechanism in dependence of GSH/GSSH ratio. Under an increase in GSH/GSSG, Grx can catalyze the deglutathionylation of proteins. The glutathionylated sulfur moiety of the protein–SSG is attacked by the thiolate anion of the enzyme (Grx-S^−^), forming the covalent enzyme intermediate (GRx–SSG) and releasing the reduced protein–SH as the first product (1). The second rate-determining step involves the reduction of Grx–SSG by GSH to produce glutathione disulfide (GSSG) as the second product, recycling the reduced enzyme (Grx–S^−^) (2). Under conditions of decreased GSH/GSSG ratio, Grx can catalyze S-glutathionylation of proteins. The S-glutathionylated Grx (Grx–SSG), formed in reaction with GSSG (3), reacts with a protein to create S-glutathionylated protein (protein–SSG) (4).

**Figure 2 molecules-26-00435-f002:**
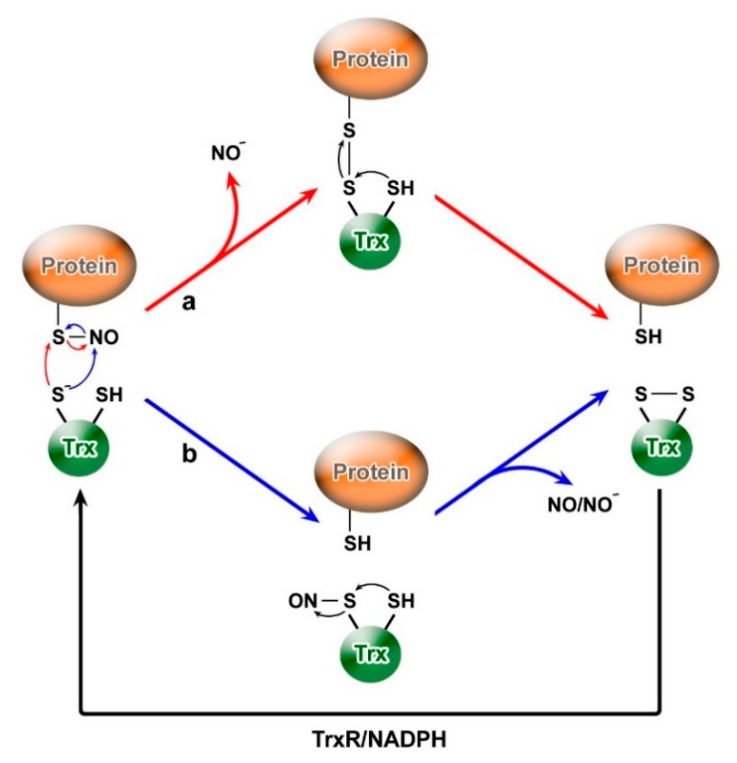
Protein denitrosylation by the Trx/TrxR system. Two thioredoxin-dependent mechanisms of protein denitrosylation are proposed: (a) formation of Trx linkage with substrate protein by a disulfide bridge; (b) trans-nitrosylation due to transient S-nitrosylation of Trx.

**Table 1 molecules-26-00435-t001:** Enzymes with capacity to catalyze S-glutathionylation and deglutathionylation.

Protein Modification	Enzyme	Reference
S-glutathionylation	GST P1-1	[32,33,34,35,36,37,38,39,49]
	GST M1-1	[32]
	Glo2	[42]
	Grx1	[48]
	Grx2	[47,48]
Deglutathionylation.	Grx1	[48,50,51]
	Grx2	[47,48,50,52]
	GSTO1-1	[53,54,55]
	Srx	[13,56,57,58]
	Trx	[59,60,61,62]
	PDIs	[63]

**Table 2 molecules-26-00435-t002:** Transnitrosylases and their substrates.

Type	Transnitrosylase	Substrate	Reference
Me-to-Cys	Cytochrome *c*	GSH	[84]
	Ceruloplasmin	GSH	[85]
		Glypican-1	[86]
	Hemoglobin	Auto-S-nitrosylation	[87]
Cys-to-Cys	GAPDH	SIRT1, HDAC2, DNA-PK	[88]
		Hsp60, Acetoacetyl-CoA thiolase, VDAC1	[89]
	Trx1	Caspase 3	[90]
		Prx1	[91]
	Caspase 3	XIAP	[92]
		Trx1	[93]
	Cyclin-dependent kinase 5	Dynamin-related protein 1	[94]
	Protein Deglycase DJ-1	PTEN	[95]
	Hemoglobin	Anion-exchanger 1 protein	[96]

**Table 3 molecules-26-00435-t003:** The major denitrosylases and their substrates.

Denitrosylase	Substrate	Reference
GSNO reductase	G-protein-coupled receptor kinase 2	[110]
	β-arrestin 2	[111]
	HIF1α	[112]
	Ras	[113]
	Ryanodine receptor 2	[114]
	Connexin	[115]
	AGT	[116]
	Dynamin-related protein 1	[117]
	Parkin	[117,118]
Thioredoxin	Caspase 3	[93,119,120]
	Caspase 9, PTP-1B, GAPDH	[93]
	NSF	[121]
	Insulin receptor, Akt, PDE3B	[122]
	Actin	[123]
	NF-κB	[124]
	NOS1	[125]
	NOS2, MEK1	[126]
	CD95, NOS3	[127]
Glutaredoxin 1	L-Cys-SNO, GSNO, Caspase 3, Cathepsin B	[103]
Protein disulfide isomerase	GSNO	[128,129,130]
Glutathione peroxidase	GSNO	[131]
Carbonyl reductase	GSNO	[132]
Ceruloplasmin	Glypican-1	[86]

## Data Availability

Data is contained within the article.

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
