# Peer review of "Glutathione in Protein Redox Modulation through S-Glutathionylation and S-Nitrosylation"

_molecules, 2021, doi:10.3390/molecules26020435_

Round 1

Reviewer 1 Report

This paper provides an overview of the redox regulation of Cys residues in both GSH and on proteins by S-glutathionylation and nitrosylation. This is an area which has been reviewed multiple times recently and although this current manuscript has some merit, it does not provide a huge advance on what is already known.

The manuscript is long and could be significantly shortened by considering general cases rather than repeating information. Thus many of the reactions of the Cys in GSH and on proteins are similar, so these could be described together and then exceptions discussed. It should be easy to cut the length of the manuscript by at least 25%.

There are a number of sections in the text that make definitive statements with few or no references. This should be corrected.

The English grammar needs some attention as there are a significant number of sentences that are not well constructed. There are also a number of spelling / typographical errors that should be corrected.

The authors should avoid the use of the general terms 'ROS' and 'oxidative stress'. These are poor terminology as they do not define what the species and systems are. Please be specific.

The authors use a number of vague terms in places (e.g. 'very high levels'). This should be avoided, as it is not at all clear what this means. Please give concentrations or similar. This is important for an understanding of the significance of these reactions. Are these reactions 1%, 10%, 50%, 100% ? - this is just not clear.

There are some areas where additional specific information should be included, such as the description of the hydroxyl amino radical (what is the structure of this exactly) and how this is formed.

Author Response

Dear reviewer. Thank you very much for the valuable comments and suggestions that we used for the correction of the manuscript.

In according to your recommendations, the manuscript has been reduced by 25%, and references to the articles of the last two years have been added.

Added more specific information related to the use of terms “ROS, oxidative stress, very high levels” (p. 4, 7, 12).

Correction was made in the structure of the hydroxyl amino radical with the description of the possible conditions for its formation (p. 14).

Thank you again for your valuable comments.

Reviewer 2 Report

This is an interesting review that summarizes the state of the art of protein S-glutathionylation and S-nitrosylation. The strength of this review is its focus on the mechanistic details of these modifications and the players involved, and this is done in a comprehensive, didactic and well-structured manner.

Major point:

The abstract promises some information about “being important for the regulation of the functional activity of proteins and intracellular processes”. The word “regulation” is also found in the title. Yet, the information concerning their regulatory role is dispersed throughout the manuscript, likely not comprehensive and the text in this context is dotted with “maybe”. References bolstering this regulatory role are frequently either isolated findings (see Ref. 18, that is the only citation for the strong sentence in the introduction (line 68ff: “Modulation of activity of these reactions in response to a change in the GSH/GSSG ratio as a result of an increase or decrease in ROSN level can make a significant contribution in the cell functional adaptation to redox changes of the environment [18].” ) or sometimes rather old (see for instance Refs 18-21). As an interested non-expert I do not (but would like to) know how solid the evidence for a regulatory role of protein S-glutathionylation and S-nitrosylation really is in the cellular context. On the systemic level, I know this is important for vasodilatation. Yet, on the cellular level I find myself asking in how far these modifications are really regulatory signals, or rather just protective (S-glutathionylation), accidental (S-nitrosylation), or even intended to inflict random damage (as is the case for instance for macrophages). This is the same with sulphur modifications which at some point an expert in the field referred to as “involved in everything”. Therefore the manuscript would gain considerably, if the authors would treat the topic of “regulation” in a similar comprehensive way as their parts on mechanism. Separated chapters for GSH and NO would be best and tables summarizing those modification on proteins that are accepted in the field as bona fide “regulatory” would be very helpful for the general reader in order to judge how important/well studied the aspect of regulation really is. These chapters do not need to be long. Further, the comprhensive listing of current state of the art reviews from the last couple of years dealing with this topic would be very helpful for the non-expert.

Minor points:

Page 1: Introduction: I find the first sentence very confusing. “Redox-dependent processes largely determine the cell viability through participating in the regulation of respiration division, bioenergetics, and programmed death.”. From my point of view, respiration is a redox-dependent process and bioenergetics is a subject. Further, line 33: GSH is also found in all other forms of life.

Page 6 and Fig. 1: The authors talk about monothiol and dithiol mechanism of protein-S-de-glutathionylation. I vaguely remember that a rather complicated “monothiol” mechanism was previously suggested for monothiol Grxs that have only one cysteine in their active site. This can be confused. The authors should clarify this and actively mentioning the existence of monothiol Grxs. They might mention the recent findings that monothiol Gxs most likely do not catalyse protein-deglutathionylation by a monothiol mechanism for structural reasons.

Page 11, line 363 a “S” seems to be missing: Pr-SNO.

Page 12 and Fig. 4: As an expert, I am sceptical whether dinitrosyl iron complexes on proteins can be repaired in the cell without being completely removed first. In any case, the indicated generalised role for a cysteine desulfurase in repairing dinitrosyl iron complexes on damaged proteins can be most likely be excluded as this enzyme has only one target protein in living cells, ISCU2. The cited rather old literature reports is likely an in vitro artefact.

Page 19 and Fig. 7: Is NO release an accepted enzymatic function of cytochrome c? There is only 1 original finding in the literature list. If this is really the entire evidence, it should not be highlighted in a figure. Furthermore, I wonder what this novel enzymatic function might be good for.

Author Response

Dear reviewer. Thank you very much for the valuable comments and suggestions that we used for the correction of the manuscript.

Considering your comment, we have tried to focus the discussion on the role of GSH/GSSG in the modulation of S-glutathionylation and S-nitrosylation and their relationship for maintenance of the cell viability. This point of view seems relevant, given the important role of GSH/GSSG and enzymes of Trx Family, which activity depends on the GSH/GSSG level, in thiol-disulfide exchange and redox-dependent processes.x

In according to your recommendations, the manuscript has been reduced by 25%, and references to the articles of the last two years have been added.

Corrections were made in the introduction (p. 1) and the section “Protein S-nitrosylation and Denitrosylation (formula, p. 8).

The sections on dinitrosyl iron complexes and role of cytochrome c in GSNO synthesis are excluded from the review.

Thank you again for your the valuable comments.

Round 2

Reviewer 1 Report

The English in the manuscript is poor and the paper needs extensive editing to make it easier to read and understand. There are a number of sentences where it is difficult to understand the sentences. 

The manuscript contains a number of errors and the authors have not responded to some of the previous comments about what 'ROS' give rise to the glutathionylation of proteins. 

The comment that neutral thiols being 'practically non-reactive' (line 48) is incorrect - this is dependent on the oxidant involved (see previous comment). Neutral thiols still show high reactivity with some oxidants.

Line 112: The description of micromolar concentrations of H2O2 reacting rapidly with protein-SH groups is incorrect. In most cases this reaction is very slow (rate constant, k, less than 100 M-1 s-1). Only in the case of specific H2O2 removing enzymes (e.g. peroxiredoxins) is this reaction fast.

There is still a degree of repetition which should be removed. See, for example, lines 277-284 and lines 448-457. Just make the target a generic species (X-H where X = GS or ProteinS).

Figure 1 could be readily simplified: the enzymatic reactions are, by definition, reversible processes, so the left and right hand side reactions are just the reverse of each other. Simply make all the arrows double headed and remove half of them.

Line 299: the nitroso group is not NO-  

The authors, when they mention specific residues, need to specify which species the enzyme comes from - these numbers vary with species (e.g. the catalytic Cys in human GAPDH is Cys152 (see line 330).

Line 341: acetyl CoA

Author Response

Dear Reviewer,

We sincerely apologize for the incomplete implementation of your recommendations in the first version of your review. Thank you again for your valuable suggestions and comments, according to which we additionally made the following corrections:

In according to your recommendations, the manuscript has been corrected in English by MDPI English Editing Service.

According to your proposal to explain "what 'ROS' give rise to the glutathionylation of proteins" we would like to clarify that we meant H2O2, which can oxidize Pr-SH to form sulfenic acid and its subsequent conversion to Pr-SSG. After correction this is presented on the lines 104-105:

“S-glutathionylation can occur when Pr-SH or GSH reacts with an oxidized derivative of the protein cysteine residue, for example, sulfenic acid (-SOH), thiyl radical (-S), S-nitrosyl (-SNO) group. Thus, when Pr-SH is oxidized (for example, by H2O2), sulfenic acid (Pr-SOH) is formed and then quickly reacts with GSH to form Pr-SSG”.

If we talk about the impact of RONS in general, then PSH may be also modified by GSNO to form PSNO and/or PSSG. This is discussed in section 4.

line 48. We agree with your remark “that neutral thiols being 'practically non-reactive' (line 48) is incorrect” and a corresponding correction has been made on the lines 46-47:

from “The pKa value of most SH-groups of cellular proteins is more than 8.0, which keeps thiol groups predominantly protonated and practically non-reactive at physiological pH values (pH 7.0-7.4).” to “The pKa value of most SH-groups of cellular proteins is more than 8.0, which keeps thiol groups predominantly protonated at physiological pH values (pH 7.0-7.4).”

line 112: We agree with your remark “The description of micromolar concentrations of H2O2 reacting rapidly with protein-SH groups is incorrect” and a corresponding correction was made:

from: “The oxidation of protein SH-groups by micromolar concentrations of H2O2 proceeds rapidly: sulfenic acid is unstable and can undergo further oxidation to sulfinic acid (Pr-SO2H) and eventually to sulfonic acid (Pr-SO3H), the formation of which, as a rule, leads to irreversible deactivation of the protein.” to “ Sulfenic acid is unstable and can undergo further oxidation to sulfinic acid (Pr-SO2H) and eventually to sulfonic acid (Pr-SO3H), the formation of which, as a rule, leads to irreversible deactivation of the protein.”

We agree with your remark “there is a repetition on lines 277-284 and 448-457”. Lines 448-457 were deleted and the sentence “The formation of GSNO can occur by the same way” has been added on lines 283-284.

We would like to leave Fig. 1 unchanged, since it can be used to trace in detail the change in the direction of reactions depending on an increase or, conversely, a decrease in the GSH/GSSG ratio.

line 299: An annoying error has been deleted thanks to your remark “the nitroso group is not NO-” and the corresponding correction has been made on the line 296:

from “transfers the nitroso group (ON-) to the protein cysteine residue” to “transfers NO to the protein cysteine residue”.

We agree with your remark “The authors, when they mention specific residues, need to specify which species the enzyme comes from” and the corresponding corrections have been made:

line 134: As a result of human Prx6 peroxidase activity, the Cys47 in active center is oxidized.

line 144-146: S-glutathionylation occurs at the Cys299 and Cys304 residues and causes conformational changes that activate the kinase activity of human AMPK.

line 198: the active site Cys32 in human GSTO1-1

line 223-224: human Prx1 can be S-glutathionylated at three out of four cysteine residues -Cys52, Cys173, and Cys83.

line 249:  In the human NOS3 reductase domain, the sites of S-glutathionylation are Cys689 and Cys908.

line 330: Human GAPDH is S-nitrosylated at the catalytic residue Cys152.

line 385: cysteine residue in the active site (Cys32 of human Trx1).

line 388-389:  the second cysteine of the Trx active center (Cys35 of human Trx1).

line 509: S-nitrosylation of human STAT3 at the Cys259 residue.

line 519: S-nitrosylation of human Prx2 at Cys51 and Cys172,

Line 340: The typing error has been deleted: “aceti-CoA thiolase” has been corrected to “acetyl CoA thiolase”.

Thank you very much again for your valuable suggestions and comments.